depression; major depressive disorder; prevention; low- and middle income countries; randomized controlled trials; meta-analysis; universal prevention; selective prevention; indicated prevention

**Corresponding author:**
Pim Cuijpers;
Email: p.cuijpers@vu.nl

# Preventing the onset of depressive disorders in low-and middle-income countries: An overview

Pim Cuijpers

Department of Clinical, Neuro and Developmental Psychology, Amsterdam Public Health Research Institute, Vrije Universiteit Amsterdam, Amsterdam, The Netherlands and International Institute for Psychotherapy, Babeş-Bolyai University, Cluj-Napoca, Romania

## Abstract

Depressive disorders constitute an important and costly public health problem and worldwide most of the disease burden is suffered in low-and middle-income countries (LMICs). Treatments only have limited possibilities to reduce the disease burden of depressive disorders. Prevention may be one of the alternative ways to further reduce the disease burden of depressive disorders. In this paper, the results of a subgroup analysis of a previous meta-analysis on the effect of preventive interventions on the incidence of depressive disorders was undertaken. Only 6% of all trials examining the possibility to prevent the onset of major depression have been conducted in LMICs, and these studies find significantly smaller effects than those in high-income settings. It is too early, therefore, to consider implementing and disseminating preventive interventions in LMICS. However, in optimal conditions and assuming that evidence-based preventive interventions will be developed, investments should be made into treatment, universal, selective and indicated prevention, as well as in social institutions focusing on larger risk factors for mental health problems.

## Impact statement

Prevention of depressive disorders is highly relevant from a public health perspective, but almost all evidence comes from high-income settings. Although selective and indicated preventive interventions have been found to be effective in high-income settings, insufficient evidence is available for low- and middle-income countries. More research into the possibilities to prevent depressive disorders in low- and middle-income countries is very much needed.

## Introduction

Depressive disorders are highly prevalent, disabling, and costly and are associated with considerably diminished quality of life, role functioning, medical comorbidity, and mortality (Whiteford et al., 2013; Cuijpers et al., 2014; Herrman et al., 2022; WHO, 2022). It has been estimated that about 280 million people worldwide suffer from depressive disorders, making this the second leading cause of global years lived with disability (WHO, 2022). Apart from the personal suffering of patients and their relatives, the economic costs are enormous, with an estimated 12 billion productive workdays lost every year to depressive and anxiety disorders, at a cost of nearly US$1 trillion (Chisholm et al., 2016). More than 80% of people with mental disorders live in low- and middle-income countries (LMICs). For example, there are more people with depressive disorders in China than the total population of Spain, and more adolescents with these disorders in India than the total population of the Netherlands. Depressive disorders affect more women than men and are also highly prevalent in vulnerable groups, such as young people, older adults, and minorities (WHO, 2022).

Preventive interventions are commonly defined as those conducted before individuals meet the formal criteria of a depressive disorder as outlined in the Diagnostic and Statistical Manual of Mental Disorders, fourth edition (DSM-IV; Mrazek and Haggerty, 1994; Institute of Medicine, 2009; National Academies of Sciences, Engineering, and Medicine, 2019). Acute and maintenance treatments are designed for those with diagnosed disorders, whereas mental health promotion centers on overall well-being not just averting illness or pathology. Prevention approaches can be divided into three categories: universal prevention that targets entire populations, regardless of whether that have an increased risk to develop a disorder; selective prevention that focuses on groups at-risk for depressive disorders; and indicated prevention directed at individuals who already have some symptoms of a mental disorder but do not meet diagnostic criteria. Preventing depressive disorders is essential for various reasons. First, it has been established that depressive disorders lead to a diminished quality of life for those affected, as well as their families. These disorders are also associated with an increased risk of early mortality

(Cuijpers et al., 2014) and general medical disorders such as diabetes and heart disease (Herrman et al., 2022). There is also a heavy economic burden associated with these conditions. Furthermore, the incidence of depressive disorders is alarmingly high - 48% of patients who have experienced a depressive episode in the past year reported it was their first time (Bijl et al., 2002). The economic costs of depressive disorders associated with incident cases are also very high (Smit et al., 2006). Although mental health care is mainly aimed at treating patients with depressive disorders, hardly anything is done about the vast influx of new cases who develop such a disorder for the first time in their lives. Currently, mental health care mainly treats patients with depressive disorders, while hardly anything is done about the vast number of new cases who develop it for the first time.

Preventing depressive disorders is also essential, since current treatments can only help to a certain degree. A modeling study in Australia estimated that up to 23% of the disease burden of depressive disorders could be lessened if all patients received evidence-based treatment – as opposed to the 16% now being achieved (Andrews et al., 2004). Moreover, approximately 40% of people with a depressive disorder go without any treatment at all. If an evidence-based treatment could be delivered to all patients with depressive disorder, a total of 34% of the disease burden could be taken away. Although this research was conducted in one country and some time ago, it still shows how current treatments can only address a limited part of the disease burden. Therefore, alternative methods such as prevention may be more effective in reducing the remaining 66% of the disease burden of depressive disorders (Andrews et al., 2004). Evidence-based treatments are not widely available in LMICs, meaning that a much smaller part of the disease burden is averted by current treatments. When building up an infrastructure for mental health care in any community, it is currently not clear whether the focus should exclusively be aimed at treatment, with the limited possibilities to reduce the disease burden, or at an integral system in which preventive services also have a place.

Each of the three types of prevention has its own advantages and challenges, so considering the implementation and dissemination of prevention should be considered separately for universal, selective, and indicated prevention. Universal prevention, aimed at a full population, regardless of their risk status, has several important advantages. The most important advantage of universal prevention is that stigma is low because everyone gets the intervention and those with a disorder or symptoms are not identified and selected for the intervention. Because they are conducted in specific settings, for example, schools, universities, and the workplace, these interventions can be integrated into existing broader programs, like well-being programs for employees or life skills programs for students.

However, there are also important disadvantages of universal prevention programs for depression. One important disadvantage is that it is very difficult to show in randomized trials that such programs actually prevent the onset of depressive disorders. To show such effects, very large randomized trials are needed with expensive diagnostic interviews (Cuijpers, 2003). Furthermore, the effects of universal interventions at the symptom level are usually small. For example, in school-based programs, a small standardized mean difference for depression was found to be 0.17 in the short term and 0.10 in the longer term (Werner-Seidler et al., 2021). Although this was significant, the effect is very small, and it is not clear whether this has a clinical meaning. In addition, the effects of

universal prevention are often realized in participants who already have baseline symptoms and should be considered a more indirect treatment instead of prevention (Cuijpers, 2022). Such effects are still relevant from a clinical perspective but should not be considered actual prevention.

Selective interventions aimed at high-risk groups include, for example, programs aimed at preventing child abuse (Park and Kim, 2017; Walsh et al., 2018) and support programs for caregivers of frail elderly (Cheng et al., 2020) or caregivers of relatives of patients with mental disorders (Chien et al., 2018). Most of these interventions do not focus directly on preventing the onset of depressive disorders, although there are exceptions like programs that aim at preventing depression in children of parents with depression (Siegenthaler et al., 2012). One important strength of these approaches is that the interventions can be adapted very well to the needs of the at-risk populations and step away from a narrow focus on mental disorders but also target daily problems that may be as important as symptoms of mental health problems in these populations. One important challenge is that trials examining these interventions usually do not examine the effects on the incidence of mental disorders. Another challenge of selective interventions is the low predictive value of most known risk factors for mental disorders (Cuijpers et al., 2021). That means that despite the increased risk, the large majority of people in a high-risk group will not develop a depressive disorder. This again complicates research examining the effects of such interventions on incidence.

The main strength of indicated preventive interventions is that the effects on the incidence of mental disorders can be examined relatively well in randomized trials. Such trials still need much larger sample sizes than treatment studies (Cuijpers, 2003), and they do require diagnostic interviews with all participants at baseline and follow-up, but they are still easier and logistically less challenging than trials on universal and selective prevention. It should therefore not come as a surprise that the evidence supporting these interventions is stronger than the supporting evidence for universal and selective prevention. It is well-established that indicated prevention indeed has a significant effect on the incidence of depression, although as we saw this may not be true for LMICs. Another strength of indicated prevention is that it is relatively easy to identify people with subthreshold depression through screening.

An important challenge for indicated prevention is that the uptake is often low. It has been estimated, for example, that only 1% of the people with subthreshold depression participated in available indicated interventions for the prevention of depressive disorders (Cuijpers et al., 2010). One important reason for this low uptake is the unwillingness of potential participants to participate. Participants are often not willing to participate because of stigma, the belief that interventions are not effective and becaue they do not consider themselves as having depressive symptoms. When the uptake is so low, these interventions may be effective for participating individuals, but the impact on public health is limited. The most important advantages and challenges of the three types of prevention are summarized in Table 1.

Treatment of existing depressive disorders is usually considered to be the most important strategy to reduce its disease burden. However, is it also possible to prevent the onset of depressive disorders, so that treatment is not needed? If services in LMICs are built up, should they focus only on treatment because resources are limited, or should they also focus on prevention? If prevention is an option, how would that look like? These are all questions that have not yet been fully answered.

**Table 1.** Advantages and disadvantages of universal, selective, and indicated prevention: An overview

| Intervention | Advantages | Disadvantages |
|---|---|---|
| Universal | | |
| | • May shift the "normal curve" of mental health toward more health | • Effects on incidence can only be examined in large and expensive trials |
| | • No stigma, because everyone gets the intervention | • Small benefit for individuals |
| | • Even small effects can have a large impact, because the whole population is reached | • High risk of bias (low quality) makes outcomes uncertain and maybe result in non-significant outcomes and no impact |
| Selective | | |
| | Can be adapted easily to the needs of the target groups | Preventing the onset of mental disorders is hardly examined, because it is typically not the primary outcome |
| | No narrow focus on mental disorders | Low specificity of most risk factors |
| | Many high-risk groups are known from epidemiological research | Low incidence rates of mental disorders |
| Indicated | | |
| | Evidence-base for preventive interventions is strong | Low uptake of many interventions |
| | Relatively easy identification through screening | Stigma |
| | | Small impact on population |

The aim of this review paper was to examine the evidence of whether prevention of depressive disorders is effective and feasible in LMICs.

## Methods

Given that there were not enough trials in LMICs to conduct a full meta-analytic review, a subgroup analysis of the few trials in LMICs from the previously mentioned meta-analysis (Cuijpers et al., 2021) was undertaken. We conducted a subgroup analysis (using a mixed-effects model) to compare the effects found in studies in LMICs compared with those found in other countries (not reported in the original meta-analysis). Further, we conducted a narrative review of prevention trials in LMICs, which only examined the effects of prevention interventions at the symptom level.

## Results

It has long been thought that it is not possible to prevent the onset of depressive disorders, because the causal pathways leading to it are unknown (Lobel and Hirschfeld, 1984; Muñoz, 1993). However, since the mid-1990s a growing number of randomized trials have started to examine the effects of preventive interventions on the incidence of depressive disorders. Although several hundreds of trials have examined the effects of interventions that aimed at prevention, by far the majority of these trials only examined the effects on levels of symptoms of these interventions, but failed to examine the reduction of new cases of depressive disorders.

In a recent meta-analysis, we examined the effects of randomized trials on the incidence of new cases of depressive disorders (Cuijpers et al., 2021). Participants were excluded if they met the criteria for a depressive disorder at baseline according to a diagnostic interview. At follow-up, another diagnostic interview was conducted to examine how many participants had developed a depressive disorder, compared with the participants in the control conditions who had not received the intervention. We included 50 randomized trials with a total of 14,665 participants. The psychological interventions were mostly based on cognitive behavioral interventions. Thirty-three were indicated interventions, 16 were selective interventions, and one was a universal intervention. The studies aimed at a broad range of different target populations, including adolescents, college students, pregnant women, and patients with general medical disorders. One year after the preventive interventions, the relative risk of developing a depressive disorder was RR = 0.81 (95% CI: 0.72–0.91), indicating that those who had received the intervention had 19% less chance to develop a depressive disorder. The effects were comparable for indicated (RR = 0.81) and selective interventions (RR = 0.79).

None of the 50 randomized trials was conducted in a low-income country and only three were conducted in middle-income countries: two in China (Zhang et al., 2014; Wong et al., 2018) and one in India (Dias et al., 2019). All 47 other studies were conducted in high-income countries. Selected characteristics of these three studies are presented in Table 2.

We conducted a subgroup analysis (using a mixed-effects model) to compare the effects found in studies in LMICs as compared with those found in other countries (not reported in the original meta-analysis) and found that studies in LMICs were significantly less effective ($p = 0.03$). The pooled RR for the three studies in LMICs was RR = 0.99 (95% confidence interval [CI]: 0.72; 1.35, $p = 0.86$; $I^2 = 0$; 95% CI: 0; 90; prediction interval: 0.16; 6.17), which was not significant. The forest plot of the effect sizes is given in Figure 1. The pooled RR of the 47 other studies that were not conducted in LMICs was RR = 0.80 (95% CI: 0.71; 0.90, $p < 0.001$; $I^2 = 38$; 11; 56; prediction interval: 0.33; 1.92). Because of the small number of trials in LMICs, we did not conduct any additional analyses. For further details of the methods and outcomes, we refer to the main paper on the meta-analysis (Cuijpers et al., 2021).

The number of studies in LMICs is too small yet to give an indication of whether preventive interventions in these countries can be effective. Also, the finding that this group of three studies pointed at significantly smaller effect sizes than studies in high-

**Table 2.** Selected characteristics of included studies[a]

| Study | Target group | Age group | Mean age | Prop. women | Intervention | Format | N sessions | country | ctr | RoB | SG | AC | BA | ITT |
|---|---|---|---|---|---|---|---|---|---|---|---|---|---|---|
| Dias et al. (2019) | Older adults | Older adults | 70 | 0.63 | CBT plus problem-solving | Individual | 6 | India | cau | low | + | + | + | + |
| Wong et al. (2018) | Primary care patients | Middle-aged adults | 54 | 0.93 | Acceptance and commitment plus mindfulness-based therapy | Group | 8 | China | cau | low | + | + | + | + |
| Zhang et al. (2014) | Primary care patients | Middle-aged adults | 49 | 0.74 | Stepped care | Individual | | China | cau | low | + | + | + | + |

cau, care-as-usual; cbt, cognitive-behavioral therapy; ctr, control group; prop, proportion; RoB, risk of bias.
[a]All three studies indicate prevention (participants had subthreshold depression but no major depressive disorder).

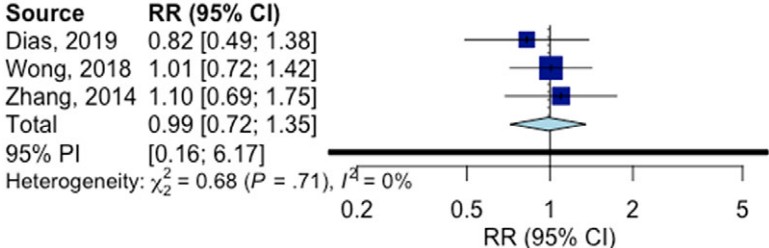

**Figure 1.** Forest plot of prevention trials conducted in LMICs.

income countries should be considered with caution because of low power. Subgroup analyses are notoriously underpowered (Cuijpers et al., 2021) and characteristics that were not measured may very well have influenced these outcomes considerably. Significant findings such as we found can therefore in no way be considered as causal evidence.

In the meta-analysis described, we only focused on trials in which participants with diagnosed depressive disorder were excluded and the incidence at follow-up was also established with a diagnostic interview. However, there are many more prevention trials, in which no (expensive) diagnostic interviews were conducted but examined the effects at the symptom level. Unfortunately, by far the majority of these trials have been conducted in high-income countries. For example, a large meta-analysis of school-based programs aimed at depression and anxiety included 118 randomized trials (Werner-Seidler et al., 2021), but only six (5%) of these were conducted in LMICs, including one in a low-income country (Nepal; Jordans et al., 2010). The income level of the countries was established according to the World Bank classification (https://datahelpdesk.worldbank.org/knowledgebase/articles/906519-world-bank-country-and-lending-groups; accessed on December 21, 2022). In another recent meta-analysis of depression prevention programs in patients with cancer, only 1 of 18 included trials (6%) was conducted in LMICs (Zahid et al., 2020), and in yet one more meta-analysis of 28 trials on the prevention of postpartum depression, only 3 (11%) were conducted in LMICs (Dennis and Dowswell, 2013), and in a meta-analysis.

This means that the knowledge of prevention programs from LMICs is very limited and almost all evidence comes from high-income countries. Because the studies from LMICs on preventing the onset of depressive disorders suggest that the effects may be smaller in LMICs as compared with high-income countries, it cannot be recommended at this moment to implement such interventions in routine care in LMICs. However, it may be possible that when more research is conducted, effective interventions may become available.

## Discussion

Depressive disorders constitute an important and costly public health problem and worldwide most of the disease burden is suffered by LMICs (Herrman et al., 2022). Treatments only have limited possibilities to reduce the disease burden of these disorders. Prevention may be one of the alternative ways to further reduce the disease burden. A growing number of studies have examined the possibilities to prevent the onset of depressive disorders. However, we saw in this narrative review that only 6% of the trials in this field have been conducted in LMICs, and these studies find significantly smaller effects than those in high-income settings. It is too early, therefore, to consider implementing and disseminating preventive interventions in LMICS.

However, considering the enormous disease burden and economic costs of depressive disorders, it is very important to invest in interventions aimed at preventing these disorders not only in high-income countries but also in low-resourced settings. Whether or not limited resources should be spent on prevention or on treatment, assuming the availability of evidence-based interventions, is a complicated issue without a clear-cut answer. In optimal conditions, in high- and low-income settings, investments should be made into treatment, universal, selective, and indicated prevention, as well as in social institutions focusing on larger risk factors for mental health problems.

Currently, there is not sufficient evidence to disseminate prevention programs in LMICs. In high-income settings, one could consider implementing indicated prevention programs, because the evidence is strong that these interventions prevent the onset of depressive disorders in some participants. If in the future

prevention programs will be developed that do have significant effects across a series of trials conducted in LMICs, there are several considerations that must be taken into account before these programs can be implemented.

First, the relative benefits of treatment and prevention must be considered. Treatments for depressive disorders are focused on acute problems and patients who are suffering. Investing in treatment solves an urgent need with direct results and outcomes. Prevention does not solve current acute problems, but only problems that will happen in the future. That may reduce the support for investing in prevention when treatments have not been fully implemented. However, it will depend on the actual effects of the prevention programs compared to those of treatment programs, whether investing only in treatment is the most rational choice. When prevention programs are very effective in terms of reducing disease burden in the longer term, it may be more rational to (also) invest in prevention programs. For example, we found in a modeling study in the Netherlands that there is an 82% probability that scaling up preventive interventions for subclinical depression is cost-effective (given a willingness-to-pay threshold of €20,000 per QALY; Lokkerbol et al., 2021). However, from a policy perspective, it is always more difficult to invest in prevention than in acute treatment, because of the urgency of treatments. Even in high-income countries, investment in prevention is typically limited and cannot compare to investments in treatment.

Second, there is insufficient evidence that universal prevention does indeed prevent the onset of depressive disorders. However, such programs can be relatively easily built into existing systems, such as schools and the workplace. Even though they may not result in reduced incidence rates, many people would consider such programs still important from a public health perspective, for example, to reduce stigma (Clement et al., 2013) or to educate people about mental health problems in general.

Third, prevention of depressive disorders should not necessarily be done through psychological interventions. Many important risk factors for mental health problems are directly related to political issues, for example, inequality, poverty, housing, employment, opportunities for education may all be related to mental health in the longer term. Although there is no evidence from trials that changing such issues will result in better mental health, it seems obvious that improving such issues will in the long run also improve mental health.

Elsewhere we have argued that prevention of depressive disorders can only be really realized when these efforts are embedded in major social institutions, have structural funding, legal consolidation, start early in life, and simultaneously target major personal and environmental determinants and their interactions (Ormel et al., 2019). This includes addressing both poor parenting and children's maladaptive personality traits and insufficient life skills, and combining universal, selective, and indicated prevention strategies with an emphasis on universal prevention. This also implies that a choice for prevention or treatment is wrong from the beginning and that investment in treatment, as well as in universal, selective, and indicated prevention is needed to reduce the disease burden of depressive disorders. That is true for both high- and lower-income settings.

The current review has several important limitations. First, there is too little research in LMICs to conduct a quantitative meta-analysis of the available evidence. It is therefore impossible to give an indication of whether such interventions are effective in LMICs. Second, because of the lack of empirical evidence, we had to conduct a narrative review, with the risks of stressing personal views and selective use of supporting evidence. Third, we have discussed the prevention of depressive disorders in LMICs as if LMICS are one homogeneous group of countries, which obviously is not the case. Research has to focus on specific countries, settings, and populations.

Despite these limitations, this study made clear that prevention of depressive disorders is a highly relevant topic, in high- and lower-income settings, but that there is currently insufficient evidence from LMICs to recommend the dissemination of preventive interventions. More research in this area is very much needed.

**Open peer review.** To view the open peer review materials for this article, please visit http://doi.org/10.1017/gmh.2023.22.

**Data availability statement.** Data availability is not applicable to this article as no new data were created or analyzed in this study.

**Author contribution.** This paper was written by Pim Cuijpers only.

**Financial support.** This research received no specific grant from any funding agency, commercial or not-for-profit sectors.

**Competing interest.** The author declares none.

**Ethics standard.** Not relevant to the current paper.

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
