## [Reviewer Report]

*Comments to Author*: This is a welcome contribution from the author, an authority on the topic of depression and the disease burden it poses globally. The paper is well reasoned and the writing compelling. It makes a clear case for the need for evidence on the effectiveness in lower-income countries of interventions of various types with the potential to prevent depression onset. The paper justifies the complementary roles in all countries of treatment and preventive interventions and related social investments in reducing the disease burden of depression. It also describes the challenges of researching interventions of the various types in any settings globally.

The paper covers global content well in describing the relevance of the topic and in the discussion and conclusions about what is now required to redress the imbalance of research and evidence across high- and lower-income countries and settings.

I have several minor comments for the author’s consideration

1. The argument could possibly include further mention of 

a. The nature and extent of the burden on young women and men 

b. The complications of depression including the higher risk of death from suicide, and premature mortality and disability from other causes including co-morbid illnesses such as diabetes and health disease – for instance in the section on ‘Why is prevention important’ (p6)

c. Related experience in other fields of health, including heart disease, and how evidence from high-income countries can be used to expedite and shape research directed by those in lower-income settings, and the mutual gains and insights

d. Reference to the Lancet-WPA commission on depression published in 2022 may be warranted and useful to the argument, eg in the Introduction

---

## [Reviewer Report]

*Comments to Author*: - Please consider using non stigmatising language, for example saying “adolescents diagnosed with depression” instead of “depressed adolescents in India”

- The terms “depression” “depressive disorders” are used interchangeably - suggest sticking to consistent terminology. 

- Please do a thorough check for citations - for example “Prevention of the incidence of new cases of major depression may be an alternative for treatment and could perhaps reduce a part of the 66% of the disease burden which is not averted by current treatments.” What is this assertion based on?

- It is inadequate to treat LMICs as a homogenous group - the analysis in this article would be greatly enhanced by a region-specific discussion of your findings and recommendations.

- While global studies and HIC setting studies have been described at different points in the article, it would be useful to bring in the comparisons into the discussion to lend weight to the recommendations made in the article.

- The language used when describing LMICs, for example: “In LMICs only a fraction of the disease burden is averted by current treatments because there is hardly any infrastructure for mental health care and treatments are hardly available.” does not adequately reflect the circumstances (historic Western Imperialism and neo-colonialism) that have led to the inequalities and gaps in healthcare systems in LMICs. The language should be re-written to be less colonial. 

- Overall the language seems a bit informal - would suggest proof reading it to make it more academic. Sentences like these - “But when building up such an infrastructure, should the focus be on building up treatment services that can reduce one third of the disease burden, or should the focus be on an integral system in which preventive services also have a place? Later in this paper we will come back to this question.” seem to belong in a chapter or a lecture, not in an academic journal. [Please ignore this comment if the journal editors are looking at moving away from the norms around academic writing.]

---

## [Reviewer Report]

*Comments to Author*: Kindly see the recommendations of the reviewers and provide revisions or a rebuttal.

---

## [Reviewer Report]

Dear Dr. Chibanda, dr. Petersen,

Thank you very much for the opportunity to revise a revised version of this paper. I have now finished the revision and I am grateful for the suggestions of the reviewers to further improve the paper. It has certainly resulted in an improved version of the paper. Below you find the comments of the reviewers and my responses.

I look forward to your decision about the revised version of the paper,

Sincerely,

Pim Cuijpers

---

## [Reviewer Report]

*Comments to Author*: While the manuscript is much improved, it still does not read well as a manuscript. In addition to a few minor editorial issues listed below, could you kindly restructure the paper so that it follows the standard reporting approach of an introduction with the aim clearly articulated, the method used (narrative review), the results, and then the discussion.

Minor editorial issues

End of second para on p. 4 change “chapter” to “paper”

Second para p. 8, Write out 33 as it is the start of a sentence

---

## [Reviewer Report]

Revision of the paper “Preventing the onset of depressive disorders in low- and middle-income countries: an overview” (manuscript GMH-22-0206.R1) according to the points raised by the Editor and the reviewers

Handling Editor: Petersen, Inge

Comment: While the manuscript is much improved, it still does not read well as a manuscript. In addition to a few minor editorial issues listed below, could you kindly restructure the paper so that it follows the standard reporting approach of an introduction with the aim clearly articulated, the method used (narrative review), the results, and then the discussion.

Reply: Thank you. I am happy that the paper is improved. In the revised version I have added the structure of a scientific paper (Introduction, Methods, results, Discussion). I have also added a research question at the end of the introduction and a general description of the Methods. 

Comment: Minor editorial issues: End of second para on p. 4 change “chapter” to “paper”

Reply: Thank you for pointing at this error. I have changed it in the revised version.

Comment: Second para p. 8, Write out 33 as it is the start of a sentence

Reply: Thank you, I have changed this.

Comment: Please also ensure your manuscript complies with the following formatting points (a copy of our author guidelines is included for reference):

- Please include the abstract in the main text document.

Reply: The abstract is included in the main text document

Comment: - Please include an Impact Statement below the abstract (max. 300 words). This must not be a repetition of the abstract but a plain worded summary of the wider impact of the article. 

Reply: Thank you, the Impact Statement was already included in the paper below the Abstract

Comment: - Submission of graphical abstracts is encouraged for all articles to help promote their impact online. A Graphical Abstract is a single image that summarises the main findings of a paper, allowing readers to quickly gain an overview and understanding of your work. Ideally, the graphical abstract should be created independently of the figures already in the paper, but it could include a (simplified version of) an existing figure or a combination thereof. If you do not wish to include a graphical abstract please let me know. 

Reply: Unfortunately, the paper is not well suited for a graphical abstract.

Comment: - Please ensure references are correctly formatted. In text citations should follow the author and year style. When an article cited has three or more authors the style ‘Smith et al. 2013’ should be used on all occasions. At the end of the article, references should first be listed alphabetically, with a full title of each article, and the first and last pages. Journal titles should be given in full.

Reply: I believe all references are correctly formatted.

Comment: - Statements of the following are required in the main text document at the end of all articles: ‘Author Contribution Statement’, ‘Financial Support’, ‘Conflict of Interest Statement’, ‘Ethics statement’ (if appropriate), ‘Data Availability Statement’. Please see the author guidelines for further information. 

Reply: All required statements are included in the paper.

Reviewer 1

Comment: Thank you for making the changes.

Reply: I want to thank the reviewer for this comment and for reading my paper critically.

Reviewer 2

Comment: Thank you, I find the minor changes and clarifications helpful

Reply: I want to thank the reviewer for this comment and for reading my paper critically.

---

## [Reviewer Report]

*Comments to Author*: 1. The introduction ends with the following statement ‘’In this chapter, the main subjects related to these questions will be presented in a narrative review, aimed at describing what prevention is, whether it is effective, what the evidence is in LMICs, and whether it is a feasible option in LMICs.’’

the scope of the paper itself should replace the final paragraph of the introduction. This reads as the introduction to a book and should introduce the paper.

2. The methods section currently reads as:

‘’ In this study, a narrative review methodology was used. Main questions related to prevention of depression were answered by giving an overview of the questions and the relevant research into these questions. First, the question was answered why prevention is important. Then an overview was presented on all randomized prevention trials in depression by summarizing a previous meta-analysis. The (few) trials in LMICs were summarized separately. In the following section a narrative overview was given of the advantages and disadvantages of the three main types of prevention: universal, selective, and indicated prevention. Finally, an overview was presented of the arguments to focus on prevention or on treatment in low-resourced settings.’’

Instead, we suggest instead you refer include the text you later use in the results of the paper to describe the approach of the systematic review you have conducted e.g.:

‘’Given that there were not enough trials in LMICs to conduct a full meta-analytic review, a subgroup analysis of the few trials in LMICs from the previously mentioned meta-analysis (Cuijpers et al., 2021), was undertaken. We conducted a subgroup analysis (using a mixed-effects model) to compare the effects found in studies in LMICs compared to those found in other countries (not reported in the original meta-analysis). Further, we conducted a narrative review of prevention trials in LMICs, which only examined the effects of prevention interventions at the symptom level.’’

3. Please review if the section currently in results under the subtitle ‘Why is prevention important?’ should actually be included in the introduction of your paper as this presents an overview rather than introduce new results of the systematic review.

4. Within the results section there seems to be some wider discussion on the results which may be better placed in the discussion section to conclude the findings in the paper – please review if you agree this is the case.

---

## [Reviewer Report]

Third revision of the paper “Preventing the onset of depressive disorders in low- and middle-income countries: an overview” (manuscript GMH-22-0206.R2) according to the points raised by the Handling Editor 

Comment: 1. The introduction ends with the following statement ‘’In this chapter, the main subjects related to these questions will be presented in a narrative review, aimed at describing what prevention is, whether it is effective, what the evidence is in LMICs, and whether it is a feasible option in LMICs.’’

The scope of the paper itself should replace the final paragraph of the introduction. This reads as the introduction to a book and should introduce the paper.

Reply: Thank you for raising this point. I have changed the last sentence of the Introduction as you indicated, but I do want to add that I think that it is rather artifical to turn this text into a conventional, empirical paper with Introduction, Methods, Results, Discussion. The paper has a lot of information apart from the subgroup analysis, meaning that if all the text you indicate in the points below is moved to the Introduction and Discussion, the paper has a very strange format: many pages introduction without subheadings, a one page methods and results section, and then many pages of discussion, again without subheadings. I think this will not help the reader in going through the paper, because it is an artificial structure on a narrative review that is not structured for such an empirical text.

Comment: 2. The methods section currently reads as: ‘’ In this study, a narrative review methodology was used. Main questions related to prevention of depression were answered by giving an overview of the questions and the relevant research into these questions. First, the question was answered why prevention is important. Then an overview was presented on all randomized prevention trials in depression by summarizing a previous meta-analysis. The (few) trials in LMICs were summarized separately. In the following section a narrative overview was given of the advantages and disadvantages of the three main types of prevention: universal, selective, and indicated prevention. Finally, an overview was presented of the arguments to focus on prevention or on treatment in low-resourced settings.’’

Instead, we suggest instead you refer include the text you later use in the results of the paper to describe the approach of the systematic review you have conducted e.g.:

‘’Given that there were not enough trials in LMICs to conduct a full meta-analytic review, a subgroup analysis of the few trials in LMICs from the previously mentioned meta-analysis (Cuijpers et al., 2021), was undertaken. We conducted a subgroup analysis (using a mixed-effects model) to compare the effects found in studies in LMICs compared to those found in other countries (not reported in the original meta-analysis). Further, we conducted a narrative review of prevention trials in LMICs, which only examined the effects of prevention interventions at the symptom level.’’

Reply: We have replaced the text of the Methods section with the text you suggested.

Comment: 3. Please review if the section currently in results under the subtitle ‘Why is prevention important?’ should actually be included in the introduction of your paper as this presents an overview rather than introduce new results of the systematic review.

Reply: I have moved this text to the Introduction.

Comment: 4. Within the results section there seems to be some wider discussion on the results which may be better placed in the discussion section to conclude the findings in the paper – please review if you agree this is the case.

Reply: I have moved the section on ‘prevention or treatment’ to the Discussion. It still feel artificial to force the text into the format of a conventional article (Introduction, Methods, Results, Discussion), but if this is really what you want then this is indeed a better solution.